# Complete Depletion of Daratumumab Interference in Serum Samples from Plasma Cell Myeloma Patients Improves the Detection of Endogenous M-Proteins in a Preliminary Study

**DOI:** 10.3390/diagnostics10040219

**Published:** 2020-04-14

**Authors:** Hana Vakili, Sharon Koorse Germans, Xiuhua Dong, Ankit Kansagra, Hetalkumari Patel, Alagarraju Muthukumar, Ibrahim A. Hashim

**Affiliations:** 1Department of Pathology, Clinical Chemistry, University of Texas Southwestern Medical Center, Dallas, TX 75390, USA; Hana.klassenvakili@utsouthwestern.edu (H.V.); Sharon.Germans@phhs.org (S.K.G.); Xiuhua.Dong@UTSouthwestern.edu (X.D.); 2Department of Internal Medicine, Hematology/Oncology, University of Texas Southwestern Medical Center, Dallas, TX 75390, USA; Ankit.Kansagra@UTSouthwestern.edu; 3Harold C. Simmons Comprehensive Cancer Center, University of Texas Southwestern Medical Center, Dallas, TX 75390, USA; Hetalkumari.Patel@UTSouthwestern.edu

**Keywords:** plasma cell myeloma, daratumumab, CD38, serum protein electrophoresis, immunofixation

## Abstract

Background: Therapeutic humanized IgG1 kappa monoclonal antibody (t-mAb), daratumumab (DARA) is a Food and Drug Administration approved drug for the treatment of relapsed/refractory plasma cell myeloma (PCM). DARA appears on serum protein electrophoresis (SPEP) and on serum immunofixation (sIFE) as an IgG kappa monoclonal immunoglobulin protein (M-protein), complicating the assessment of the patients’ response to therapy. A more ominous threat to patient safety can occur with the misinterpretation of the presence of a small t-mAb spike as being the residual product of the patient’s neoplastic clone, presented either as oligoclonality or new clonality, which could result in incorrect interpretation of failure to achieve remission. Methods: In this report, we describe a novel and cost-effective technique based on biotinylated recombinant CD38 and streptavidin-coated magnetic beads to capture and remove residual DARA present in PCM patient serum samples. The treated samples are then run like regular samples on SPEP and sIFE. We validated this simple technique in DARA-spiked PCM samples and patient samples on DARA treatment. Results: Our simple capture technique completely extracted DARA in all of the tested serum specimens and allowed the assessment of residual M-protein without DARA interference. The results were reproducible and highly specific for DARA, and did not have any impact on endogenous M-protein migration and quantification by SPEP and sIFE. The cost of this technique is much lower and it can be performed in-house with a very short turnaround time compared to the currently available alternative methods. There is a great need for such reflex technologies to avoid interpretation errors. Conclusions: This method is an effective way to eliminate DARA interference in SPEP and sIFE, and can be easily implemented in any clinical laboratory without any patent restriction. This simple technique can be adopted for other t-mAbs using their respective ligands and will help to reduce additional doses of toxic treatment and further testing in patients on t-mAbs with a false positive M-protein spike.

## 1. Introduction

Plasma cell myeloma (PCM) develops due to expansion of malignant plasma cells secreting a monoclonal immunoglobulin protein (M-protein). CD38 is a transmembrane glycoprotein expressed in high levels by plasma cells and it is involved in migration, via interaction with CD31 and thus controlling receptor-mediated adhesion. Under normal conditions, CD38 is expressed in varying intensities on myeloid cells, T cells, hematogones and blasts.

Normal and clonal plasma cells have high CD38 expression, which has made CD38 a specific and effective target for therapeutic antibodies targeting this cell surface molecule in malignant plasma cells [1,2]. Daratumumab (DARA) is a humanized anti-CD38 IgG1 kappa monoclonal antibody jointly developed by Genmab and Janssen Biotech (a Johnson and Johnson subsidiary) and was approved by the Food and Drug Administration in 2015 for the treatment of relapsed and refractory myeloma [2]. Clinical trials are also currently testing the efficacy of DARA in treatment of patients with high-risk monoclonal gammopathy of undetermined significance (MGUS) and low-risk smoldering plasma cell myeloma. The monoclonal antibody binds to plasma cells, which have a high expression of CD38, and causes destruction of plasma cells via cell-mediated and antibody-dependent cytotoxicity [2]. The drug remains in circulation for as long as 70 days post-treatment and can cause false positive alarming results [3].

The first report of interference due to therapeutic monoclonal antibodies was reported approximately a decade ago, when serum protein electrophoresis (SPEP) and serum immunofixation (sIFE) ordered for a PCM patient on Siltuximab (anti-IL-6 tmAb), detected as an IgG kappa component in the gamma fraction on sIFE [4]. The International Myeloma Working Group (IMWG) has established criteria for clinical response to treatment in PCM, which includes changes in serum/urine M-protein levels, determined by SPEP and sIFE [5,6]. Based on IMWG criteria, a complete response can be concluded with complete absence of serum and urine M-protein. However, DARA is detectable by SPEP and sIFE assays as an IgG Kappa component and hence can be interpreted as false positive M-protein component. This can pose a serious challenge when differentiating a residual disease M-protein and is especially true since the most common type of PCM is IgG kappa. Many laboratories report out all visible M-protein values and monoclonal bands without reference to the monoclonal protein secreted by the original malignant plasma cell clone [7]. This is a common practice at several reference laboratories, which do not have access to patient medical records. This can negatively affect the treatment outcomes as defined by complete response status. There is also a lack of education in practicing guidelines for clinicians involved in management of patients with PCM receiving humanized monoclonal therapeutic antibody therapy.

The specific aim of this article is to describe a simple and effective method to remove the circulating residual DARA from the serum that can be implemented in clinical laboratories with a short turn-around time, lower cost and with no requirement for additional instrumentation.

## 2. Materials and Methods

### 2.1. Patient Samples

This study was initiated as a quality improvement project to evaluate a new method to remove DARA interference in SPEP and sIFE tests. Necessary approval was obtained from the Institutional Review Board at the University of Texas (UT) Southwestern Medical Center (STU-2019-0740, 20 August 2019). PCM patients receiving DARA (900–1500 mg/biweekly) complementary to bortezomib/thalidomide/dexamethasone therapy with a newly-identified IgG kappa M-component, suspicious for interference, were identified by the Hematology-Oncology groups. Serum specimens from these patients, as well as patients with no monoclonal components, were stored at −80 °C till further analysis.

### 2.2. Biotinylated CD38 Specific Extraction Method

DARA (Darzalex 100 mg/5 mL—Janssen, Horsham, PA, USA) was obtained from UT Southwestern Medical Center Pharmacy for in vitro spiking and titration studies. Patient serum pools were prepared using *n* = 10 discarded serum samples with normal electrophoretic mobility and no endogenous M-protein. These were used for the initial proof of principle studies. Aliquots of pooled donor sera (20 µL) with normal electrophoretic mobility (no endogenous M-proteins, *n* = 10) were spiked with 0.5 g/L DARA. The concentration of DARA evaluated in this study was chosen to approximate five times greater than the serum Cmax values attained based on available literature on Phase 1/2 studies (DARA: 993 μg/mL after Dose 7 at 16 mg/kg). The spiked sera aliquots were then supplemented with 0.125–0.5 g/L of biotinylated recombinant, human CD38 (Sino Biological, city, country, Catalog number: 10818-H08H-B) and incubated for ten minutes on an end-over-end tube rotator at room temperature. In total, one hundred µg of Dynabeads M-270 Streptavidin (Invitrogen, city, country Catalog number 65306) was added to each mixture and was incubated for a further five minutes. The complex of dynabeads, M-270 Streptavidin beads, biotinylated CD38 and spiked DARA was separated on a magnetic stand (two minutes for an effective separation). The concentration of Streptavidin-coated Dynabeads was chosen based on the manufacturer’s recommendation. Separated serum with final volume of 80 µl (thus 1:4 dilution of neat serum) was subsequently run by electrophoresis (Sebia Hydrasys 2, Lisses, France) as per the laboratory’s standard SPEP and sIFE procedures in accordance with the manufacturer’s instructions. The schematic of this methodology is presented in Figure 1.

All three concentrations (0.125, 0.250 and 0.500 g/L) of recombinant CD38 were able to completely remove the DARA in spiked pooled sera. Based on this finding, we decided to use 0.125 g/L concentration for further evaluation in PCM patients. In order to demonstrate that the recombinant biotinylated CD38-DARA complex does not affect the endogenous M-protein migration and hence the analytical specificity of this method, sera from patients with PCM (IgG kappa, *n* = 6) who had not received DARA as a therapy were spiked with 0.5 g/L DARA, and the impact on the migration of endogenous monoclonal protein band (IgG kappa) following serum pre-treatment with CD38-labelled beads was assessed.

The impact of this pre-treatment and the efficiency of recombinant biotinylated CD38 was also tested in sera from patients with PCM who were receiving DARA as a therapy (*n* = 10). Three trained individuals in the interpretation of SPEP and sIFE results independently evaluated the gels and results of CD38-treated PCM samples.

## 3. Results

### 3.1. Extraction of Spiked Daratumumab (DARA) from Normal and Plasma Cell Myeloma (PCM) Patients’ Sera

Normal pooled patient sera spiked with 0.5 g/L DARA resulted in a clear IgG kappa M-protein on sIFE gel. Incubation of DARA-spiked serum with 0.125–0.5 g/L of biotinylated recombinant CD38 and magnetic beads successfully extracted the DARA with a magnetic stand and completely eliminated the IgG kappa component on sIFE gel with all tested concentrations of recombinant CD38 (Figure 2).

Sera from PCM patients (*n* = 6) not receiving DARA were also spiked with 0.5 g/L DARA and treated with 0.125 g/L biotinylated CD38 prior to be run on sIFE gel. The recombinant biotinylated CD38-DARA complex removal from the spiked sera of these PCM patients did not affect the endogenous M-protein migration (Figure 3). Thus, this technique successfully and specifically extracted the DARA without affecting the migration of the endogenous M-protein.

The impact of the recombinant biotinylated CD38-DARA complex on the endogenous M-protein quantification in six patients with a range of endogenous M-protein concentration (0.3–2.9 g/dL) was also assessed. The Bland–Altman difference versus the average indicated a −0.0033 g/dL bias in measurement of endogenous M-protein post-CD38 treatment. Correlation of the quantitation pre- and post-CD38 treatment indicated a significant correlation with R squared = 0.9984 (Figure 4).

### 3.2. Extraction of in Vivo DARA in PCM Patients Receiving DARA Treatment

The impact of the pre-treatment and efficiency of recombinant biotinylated CD38 was also tested in sera from patients with PCM who were receiving DARA as a therapy. CD38 ligand antibody efficiently removed DARA band in each one of the ten patient samples evaluated. The sIFEs of these ten patients, including their original M-component, are presented in Appendix A. A representative patient sample is presented in Figure 5. We also tested PCM patients with overlapping IgG kappa and receiving DARA. In all cases, CD38 treatment could not remove the endogenous IgG kappa M-Protein (Appendix A).

## 4. Discussion

The development of monoclonal antibodies such as DARA and Elotuzumab has expanded treatment options for plasma cell myeloma patients, leading to a dramatic improvement in patient outcomes [8]. There are, however, specific challenges in the assessment of response to this therapy due to an increased interference caused on sIFE gels, which are used as a first-line diagnostic and monitoring tool in PCM patients [3,4]. A false positive interpretation of the M-protein due to therapeutic monoclonal antibodies such as DARA can lead to the misinterpretation of responses to therapy in light of the detection of a new IgG kappa M-protein post-treatment. Moreover, DARA has been detected in circulation 70 to 100 days post-infusion, causing interference long after the drug has been stopped *(3).* In our study, we demonstrate the use of biotinylated recombinant CD38, which is the specific ligand for the DARA antibody, and its efficiency in removing the drug spiked in normal patient sera as well as endogenous residual circulatory DARA in sera from PCM patients.

DARA is specific for CD38 as its ligand, and thus can be used as a specific target to deplete residual circulatory DARA. This methodology does not require any instrument specification which could add to the cost. Furthermore, interpretation of sIFE gels, pre- and post-CD38 treatment, is easy and does not require any further training as the absence of IgG kappa post-DARA therapy would indicate complete depletion of DARA without affecting the endogenous M-protein migration and quantity. This is unlike the DARA-specific sIFE reflex assay (DIRA/Hydrashift 2/4 daratumumab) that is based on the use of anti-DARA antibodies to shift the migration of DARA band [9,10]. More specifically, in instances where the patient’s endogenous IgG kappa M-protein co-migrates with DARA, the result can be misinterpreted as treatment effect. The presence of multiple endogenous IgG kappa clones can be confused with a shifted DARA band on the DIRA assay. The DIRA assay, in addition, requires specific instrumentation with a higher cost, and being a separate assay requires additional time for reporting out results. It is also mostly available through reference laboratories, which increases the turnaround time significantly. The technique described in this article can however be implemented in any laboratory performing SPEP and sIFE, without the need for specific instrumentation. It can also be easily implemented by a trained clinical laboratory medical technologist with little or no training. This methodology is a very cost-effective reflex method (20% of DIRA cost) with no significant delay in the reporting of the results to facilitate evaluation of PCM patients to therapy. This reflex assay will be most useful in the assessment of PCM patients with near complete response when the original endogenous M-protein has decreased significantly and an IgG kappa reflective of the therapeutic monoclonal antibody therapy is observed on SPEP and sIFE.

Use of CD38 in our methodology to deplete the residual DARA in sera from PCM patients has similar limitations as DIRA and Hydrashift assays as they are only specific for DARA [9,10]. This limitation has become increasingly challenging as several new therapeutic monoclonal antibodies (t-mAbs) emerge in the practice of targeted medicine. However, the principle of our methodology allows for adaptation for other t-mAbs by using their respective recombinant ligands. This elimination method takes 15 minutes without additional instrumentation and can be implemented easily in any laboratory set up.

Mass spectrometry is an alternative methodology in the detection of M-proteins as well as t-mAbs [11]. DARA and other t-mAbs have been detected and distinguished to some extent using molecular mass as a measurement tool, as it provides increased specificity based on unique masses to each monoclonal antibody [12,13]. However, specificity and sensitivity of mass spectrometry can be compromised due to additional immunoglobulin extraction or other enrichment techniques [14]. Furthermore, implementation of mass spectrometry in every clinical laboratory is challenging as it requires significant technical expertise and high cost of development. This may not suit small clinical laboratories due to the economic burden on the patient population of interest. The advantage of mass spectrometry methodology is that it does not rely on antibody-specific reagents such as recombinant antigens, as described in this report and/or anti-idiotypic antibodies. However, time and expertise are a big challenge for many diagnostic laboratories.

Based on the 2018 College of American Pathologist Laboratory Challenge, a newly-identified M-protein was misinterpreted as a new clone by 47% of pathologists among 700 participants and only 53% of pathologists assumed that the new M-protein was associated with the therapy. It is highly recommended that a thorough review of the patient’s medical history be done prior to reporting any new M-protein component, which might be reflective of the therapy. Furthermore, there is no standardization in reporting SPEP and sIFE tests of PCM patients receiving therapeutic monoclonal antibodies. We recommend that an interpretative comment be included in SPEP and sIFE reports signed out with a new IgG kappa M-component, indicating that an interference can be caused by DARA. We also recommend that the patient’s chart be flagged for the drug if the patient has or is receiving it. This will prevent gaps in patient care by making diagnosticians aware that an interference is possible in laboratory tests, including not only SPEP/sIFE, but also in blood banks and in flow cytometry, where these t-mAbs can interfere with the assays and hence interpretation. Long-term studies to assess the impact on patient care/follow-up due to incorrect complete response classification are required to truly address this problem.

The simple methodology using biotinylated recombinant CD38 to deplete residual DARA can facilitate the complete response classification of PCM patients with an IgG kappa clone, which might be reflective of DARA therapy. Thus, it can prevent additional unnecessary testing which is done to follow up the patient and additional doses of cytotoxic therapy.

## Figures and Tables

**Figure 1 diagnostics-10-00219-f001:**
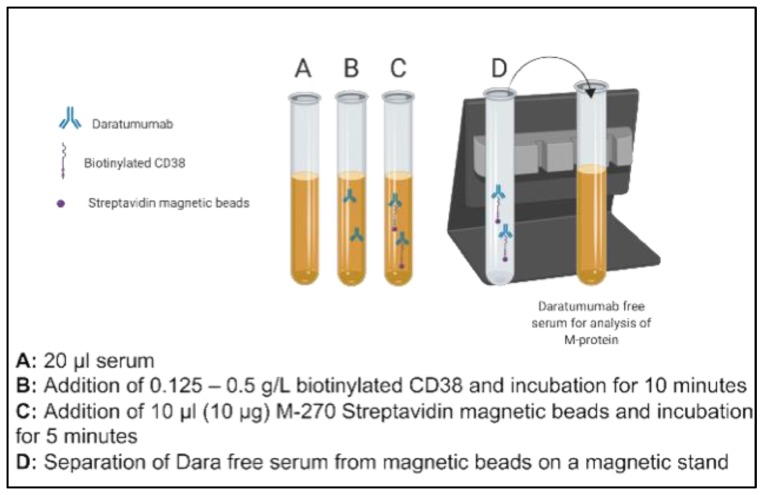
Schematic illustration of steps of a novel immunoaffinity method to deplete residual daratumumab (DARA) using biotinylated recombinant full length CD38.

**Figure 2 diagnostics-10-00219-f002:**
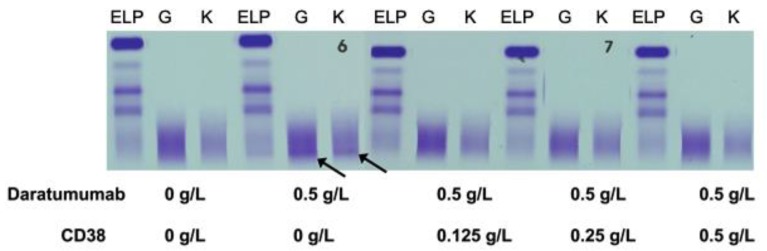
Pooled patient sera with no endogenous monoclonal immunoglobulin protein (M-protein) spiked with DARA resulted in a clear IgG (G) kappa (K) M-protein (indicated by arrows) on serum immunofixation (sIFE) gel. Incubation of DARA-spiked serum with 0.125–0.5 g/L of biotinylated recombinant CD38 and magnetic beads successfully extracted the DARA with a magnetic stand and completely eliminated the IgG kappa component on sIFE gel with all tested concentrations of recombinant CD38.

**Figure 3 diagnostics-10-00219-f003:**
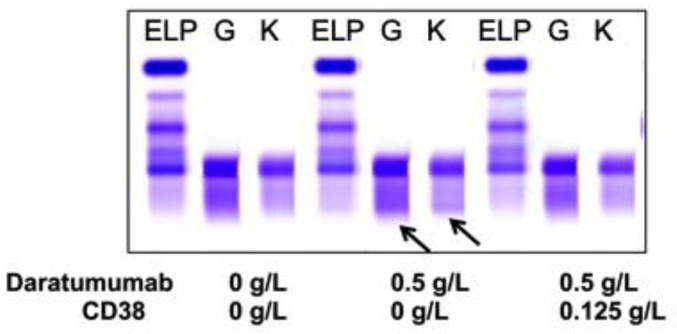
A representative of plasma cell myeloma (PCM) patient sera with endogenous IgG (G) kappa (K) M-protein spiked with DARA resulted in a clear IgG kappa M-protein (indicated by arrows) on sIFE gel. Incubation of DARA spiked PCM serum with 0.125 g/L of biotinylated recombinant CD38 and magnetic beads successfully extracted the DARA with a magnetic stand and completely eliminated the IgG kappa component on sIFE gel with all tested concentrations of recombinant CD38. CD38/DARA complex does not affect the endogenous M-protein migration.

**Figure 4 diagnostics-10-00219-f004:**
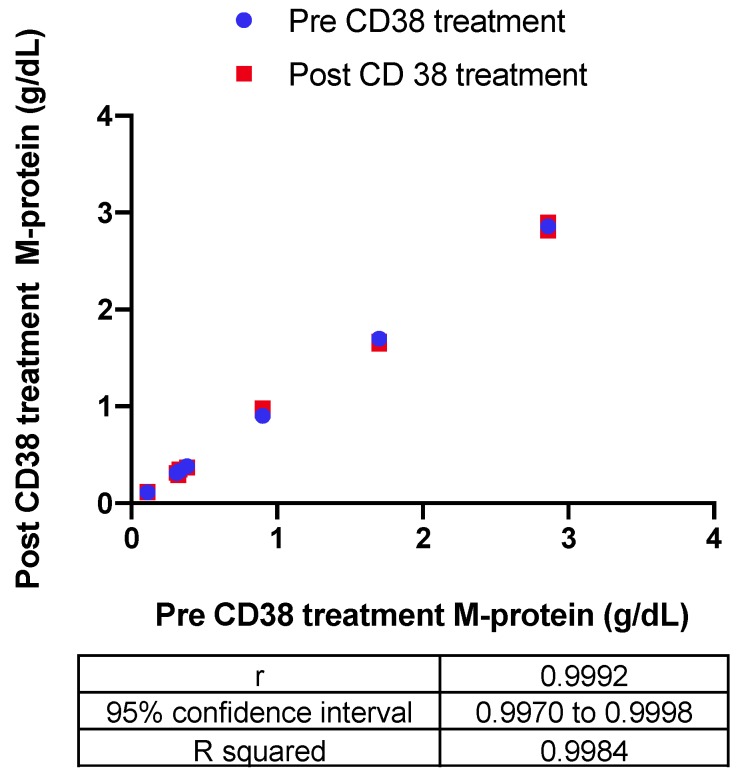
The impact of the recombinant biotinylated CD38-DARA complex on the endogenous M-protein quantification in six patients with a range of endogenous M-protein concentration (0.3–2.9 g/dL) was assessed based on the correlation of the quantitation pre- and post-CD38 treatment (R squared = 0.9984)**.**

**Figure 5 diagnostics-10-00219-f005:**
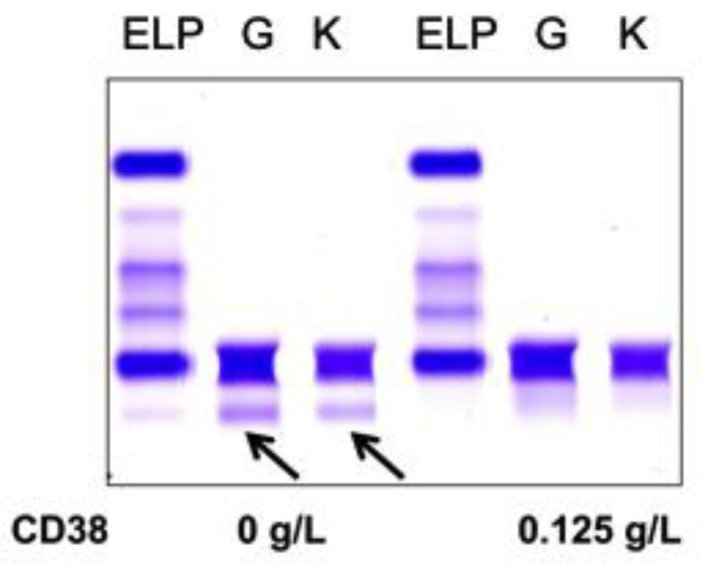
A representative patient sample is shown for illustration of the impact of pre-treatment and efficiency of recombinant biotinylated CD38 tested in sera from patients with PCM (IgG kappa indicated by arrows) who were receiving DARA as a therapy. CD38 ligand antibody completely removed the IgG kappa band associated with DARA.

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
