# Peer review of "Complete Depletion of Daratumumab Interference in Serum Samples from Plasma Cell Myeloma Patients Improves the Detection of Endogenous M-Proteins in a Preliminary Study"

_diagnostics, 2020, doi:10.3390/diagnostics10040219_

Round 1
Reviewer 1 Report
This is an interesting paper and raises some important issues surrounding DARA interference on m-protein detection. I have some comments and questions
Minor comments: Dara is abbreviated but still used in full in places..should just use abbreviated here in; Line 73 "are" should be "a"
Main comments: DARA is only used to treat refractory/relapsed myeloma and as the authors imply in the Intro a new m-protein would look suspicious - The DARA band also seems to appear in the same position. For the technique to be applied, the lab would need to be alerted the patient was on DARA. If the clinician is made aware of DARA interference (which 91.5% are according to the survey) they could just compare the IFE to the original. Is a new laboratory technique required or is this an issue for clinicians? Its hard to get a feel while in theory interference is a problem, if in practice its actually very rare? Spiked concentration is used at 5 times the expected. One could argue that it should have also been tested at the lower expected concentration to a) show it can indeed be seen clearly on IFE - in both normal and patient samples and b) demonstrate the effectiveness of the technique at the concentration that would actually be seen in serum, not just unusually high. Its particularly important to show that it is a problem at the lower levels and thus needs removing, within this study itself. Could banding be seen in all 10 patients receiving DARA? This isn't clear from the results, which focuses on single patient cases as examples. The IFE results from all patients should be shown in a panel figure or supplemental material. Only 16 patients included in study in a disease with large heterogeneity. Were all patients IgG kappa myeloma? Did the DARA band ever overlap with the patients m-protein? Again, if DARA banding is very distinct then the need for the technique could be avoided by comparing to previous IFE. There needs to be more evidence, either here or from citing previous studies, that DARA overlaps with residual m-protein The survey results need to be more clear...on those looking at IFE with and without DARA, specifically how many would be erroneously interpreted? What does it mean by only 65.8% had the knowledge for diagnosis/follow up? Are they not sure in what context the lab tests are used..how are they diagnosing/following them up then if not using criteria? Are the 44% not directly involved in patient care? More context needs to be given to understand the relevance of the findings for those who do not work in the US healthcare system The discussion makes sensible recommendations to overcome this issue -education, medical history known, interpretive comment to flag interference.Author Response
We would like to take this opportunity to thank the reviewers for their comments. We have addressed the comments in terms of changes to the text and through inclusion of additional data shown in our itemized responses. We believe our manuscript has been strengthened as a result of their comments and the review process.
Thank you for the further consideration of our manuscript.
RESPONSE TO REVIEWS:
Reviewer 1
This is an interesting paper and raises some important issues surrounding DARA interference on m-protein detection. I have some comments and questions
Minor comments:
1) Dara is abbreviated but still used in full in places..should just use abbreviated here
In the revised manuscript, DARA instead of Daratumumab has been used throughout as recommended.
2) in; Line 73 "are" should be "a"
This is corrected in the revised manuscript.
Main comments:
1) DARA is only used to treat refractory/relapsed myeloma and as the authors imply in the Intro a new m-protein would look suspicious - The DARA band also seems to appear in the same position. For the technique to be applied, the lab would need to be alerted the patient was on DARA. If the clinician is made aware of DARA interference (which 91.5% are according to the survey) they could just compare the IFE to the original. Is a new laboratory technique required or is this an issue for clinicians? Its hard to get a feel while in theory interference is a problem, if in practice its actually very rare?
We appreciate this insightful question from the reviewer. Certainly, appearance of a new band on SPEP and sIFE is a concern for both pathologists as well as clinicians and it should trigger a thorough review of the patient’s chart for possibility of a drug interference. As the reviewer suggested it would be ideal to compare the pre (original) and post-DARA IFE results. However, our clinicians felt the need for this new approach for the following reasons:
a) To have an evidence based approach rather than an assumption based comparison of pre and post results. Determining what is “pre” can be a concern when a DARA patient is followed over a longer period of time and there are changes in other regimens of drugs.
b) It is not uncommon for PCM patients to switch Clonality and therefore, just the knowledge/awareness about the migration of DARA band is not sufficient in these cases as new clone may be co-migrating with DARA band. Many of our patients undergo stem cell transplant and we have observed new clones evolving constantly, hence, an effective differentiation approach is required.
c) Although its remains possible for the clinician to compare pre- and post DARA sIFEs. This approach will save clinician time in going back and forth digging into old EMR IFE records (assuming that the patient has a previous IFE on file. We are a referral center and likely that pre-therapeutic gels may not be available). . Moreover, the way these EMR records are currently setup, side by side comparison of IFE results is not feasible.
2) Spiked concentration is used at 5 times the expected. One could argue that it should have also been tested at the lower expected concentration to a) show it can indeed be seen clearly on IFE - in both normal and patient samples and b) demonstrate the effectiveness of the technique at the concentration that would actually be seen in serum, not just unusually high. Its particularly important to show that it is a problem at the lower levels and thus needs removing, within this study itself.
We accept the reviewer’s comment. Others have tested lower doses of daratumumab and the lowest concentration observed on SPEP and sIFE was 0.2 g/L whereas 0.1 g/L was not visually obvious [1]. The maximum predicted concentration of daratumumab in serum after weekly doses is 1 g/L which is a concern [2]. However, it is not a common practice to test these patients on a weekly basis and Most often we observe 0.5 g/L concentration in our practice. Hence, we chose 0.5 g/L concentration of daratumumab spiking in both normal and PCM patients to make sure that our methodology is sufficient to remove daratumumab effectively. We also tested further lower concentrations which were not visually obvious, especially when we were scanning those images. For this reason we chose to stick with 0.5 g/L concentration of DARA.
3) Could banding be seen in all 10 patients receiving DARA? This isn't clear from the results, which focuses on single patient cases as examples. The IFE results from all patients should be shown in a panel figure or supplemental material.
Yes, the IgG kappa band related to daratumumab was observed in all 10 patients tested. As per the reviewer’s suggestion, all of the sIFEs of these patients are now included in the Supplementary Figure 1 of the revised manuscript.
4) Only 16 patients included in study in a disease with large heterogeneity. Were all patients IgG kappa myeloma?
No, they were not all IgG kappa. The M-protein type for the 10 patients tested has been now included in the Supplementary Figure 1. However, the Protein type of IgG kappa in 6 PCM patients who were not receiving daratumumab was tested for the impact of treatment on quantification of endogenous M-protein post CD-38 treatment.
5) Did the DARA band ever overlap with the patients m-protein? Again, if DARA banding is very distinct then the need for the technique could be avoided by comparing to previous IFE. There needs to be more evidence, either here or from citing previous studies, that DARA overlaps with residual m-protein
We thank the reviewer for asking this important question. Daratumumab can indeed overlap with the endogenous IgG kappa M-protein. This is in fact seen in many patients that we tested. However, we emphasized on including more non-IgG kappa M-proteins in our study to ensure that we can see the removal of daratumumab by using CD38. With this assurance, we could assume if a PCM patient with endogenous IgG kappa tested with this reflex assay would achieve the complete response criteria if CD38 would result in removal of the only IgG kappa band. If the IgG kappa post CD38 treatment persist then this can be interpreted that complete response has not achieved as shown with representative sIFEs of PCM patients with IgG kappa in Supplementary Figure 2 in the revised manuscripts.
6) The survey results need to be more clear...on those looking at IFE with and without DARA, specifically how many would be erroneously interpreted? What does it mean by only 65.8% had the knowledge for diagnosis/follow up? Are they not sure in what context the lab tests are used..how are they diagnosing/following them up then if not using criteria? Are the 44% not directly involved in patient care? More context needs to be given to understand the relevance of the findings for those who do not work in the US healthcare system
We agree that the survey results might have not been presented well in this study and we decided to remove them from the revised manuscript based on the Reviewer’s 2 suggestion.
7) The discussion makes sensible recommendations to overcome this issue -education, medical history known, interpretive comment to flag interference.
We completely agree and we have started providing comments upon chart review if there in any indication of daratumumab therapy.
References:
- McCudden, C., et al., Monitoring multiple myeloma patients treated with daratumumab: teasing out monoclonal antibody interference. Clin Chem Lab Med, 2016. 54(6): p. 1095-104.
- Overdijk, M.B., et al., Antibody-mediated phagocytosis contributes to the anti-tumor activity of the therapeutic antibody daratumumab in lymphoma and multiple myeloma. MAbs, 2015. 7(2): p. 311-21.
Reviewer 2 Report
The paper describes a new method to remove the analytical interference caused by monoclonal therapeutic antibodies in serum electrophoresis and immunofixation. The topic is certainly of interest and the study is adequately presented. In my opinion, the inclusion of the survey results in a paper describing an experimental study is not appropriate; I suggest to remove it and to present it as a separate paper (Letter to the Editor?). The manuscript content should be modified accordingly. Also, I suggest that the title includes the term “preliminary results”, since only 10 patients treated with DARA have been studied.
Specific comments.
Abstract. Line 40. The sentence seems to imply that the technique presented in the study can be used to eliminate the interference caused by other biologics besides DARA. This is not true as the Authors correctly indicate in the manuscript (see lines 240-243); please modify.
Introduction. Lines 76-77. I do not understand the reasons why the “reference laboratories” have “typically” no access to patient medical records; please clarify.
Results. Line 144: I do not understand if n=10 means that 10 samples without MC have been used to make the pool to be spiked with DARA or if 10 pools have been prepared. If this is the case, why 10 pools? From the electrophoretic patterns of fig 2, it seems that only one pool has been used of the experiment (the patterns are very similar). Fig 4 A: I see 12 circles, while the text (and the figure legend) reads that 6 samples have been used for the experiment. More information is needed for the experiment of fig 5: the range of concentration of the original MCs and of the DARA peaks would be very useful.
Discussion. Line 228-229 and 245-246. At the best of my knowledge, DIRA requires additional reagents, but the IFEs are run on the same instrument used for the routine IFEs; so, no need of additional instruments. Line 230 “It is also currently available only in reference laboratories”; DIRA is commercially available, so the meaning of the sentence is not clear to me. Line 234, it is stated that the new test costs 20% less than DIRA; in the absence of more detailed information (which is the cost of the reagents used in the present study? which is the cost of a DIRA test?), it is perhaps better not to indicate a specific value.
Author Response
We would like to take this opportunity to thank the reviewers for their comments. We have addressed the comments in terms of changes to the text and through inclusion of additional data shown in our itemized responses. We believe our manuscript has been strengthened as a result of their comments and the review process.
Thank you for the further consideration of our manuscript.
RESPONSE TO REVIEWS:
Reviewer 2:
The paper describes a new method to remove the analytical interference caused by monoclonal therapeutic antibodies in serum electrophoresis and immunofixation. The topic is certainly of interest and the study is adequately presented.
1) In my opinion, the inclusion of the survey results in a paper describing an experimental study is not appropriate; I suggest to remove it and to present it as a separate paper (Letter to the Editor?). The manuscript content should be modified accordingly.
We agree with the reviewer and have decided to remove them from the revised manuscript based on your suggestion.
2) Also, I suggest that the title includes the term “preliminary results”, since only 10 patients treated with DARA have been studied.
We agree and we have revised the title to “Complete Depletion of Daratumumab Interference in Serum Samples from Plasma Cell Myeloma Patients to Improve Detection of Endogenous M-proteins in a Preliminary Study” to reflect the small population of PCM patients tested in this study.
Specific comments.
1) Abstract. Line 40. The sentence seems to imply that the technique presented in the study can be used to eliminate the interference caused by other biologics besides DARA. This is not true as the Authors correctly indicate in the manuscript (see lines 240-243); please modify.
We agree with the reviewer and we have modified the statement to “This simple technique can be adopted for other t-mAbs using their respective ligands and will help to reduce additional doses of toxic treatment and further testing in patients on t-mAbs with a false positive M- protein spike.” in the revised manuscript.
2) Introduction. Lines 76-77. I do not understand the reasons why the “reference laboratories” have “typically” no access to patient medical records; please clarify.
We have removed Typically from the sentence. We routinely send out samples to reference laboratories. They have no access to hospital specific EPIC charts and only provide test results that are required to be interpreted by clinicians.
3) Results. Line 144: I do not understand if n=10 means that 10 samples without MC have been used to make the pool to be spiked with DARA or if 10 pools have been prepared. If this is the case, why 10 pools? From the electrophoretic patterns of fig 2, it seems that only one pool has been used of the experiment (the patterns are very similar).
We would like to clarify that we prepared only a pooled normal sera from 10 samples with no M-protein and used it for spiking with daratumumab and increasing doses of CD38 treatment as presented in Figure 2. Therefore, as you pointed out, it is expected to see a consistent similar electrophoretic pattern in this figure as it is a pooled sample used for all groups.
4) Fig 4 A: I see 12 circles, while the text (and the figure legend) reads that 6 samples have been used for the experiment.
We would like to apologize for this confusion. We have removed Figure 4A as we thought it was unnecessary to present the difference between the quantification values of M-Protein pre- and post-CD38 treatment as it is already indicated in Figure 4B graph in the correlation studies. We have corrected the text in the legend accordingly.
5) More information is needed for the experiment of fig 5: the range of concentration of the original MCs and of the DARA peaks would be very useful.
We agree with the reviewer’s input. Accordingly, more information regarding the 10 PCM patients tested are now presented in Supplementary Figure 1 of the revised manuscript. In all cases daratumumab related IgG kappa band was >0.5 g/L.
6) Discussion. Line 228-229 and 245-246. At the best of my knowledge, DIRA requires additional reagents, but the IFEs are run on the same instrument used for the routine IFEs; so, no need of additional instruments. Line 230 “It is also currently available only in reference laboratories”; DIRA is commercially available, so the meaning of the sentence is not clear to me.
We thank the reviewer for asking this important question. DIRA can only be run on Hydrasys 2 system and at present it is a send-out test to LabCorp for us. DIRA assay can also be performed in-house using the current Sebia instrument set-up.
7) Line 234, it is stated that the new test costs 20% less than DIRA; in the absence of more detailed information (which is the cost of the reagents used in the present study? which is the cost of a DIRA test?), it is perhaps better not to indicate a specific value.
Direct cost for our test was around $45. This included the cost of biotinylated CD38, Streptavidin dynabeads, DARA positive control, 12 tube magnetic separation rack, cost for SPEP and IFE, and tech time. Currently, our reference lab charges for DIRA test is around $270. Reference lab charges are confidential information and cannot be openly shared. For this reason, we have used an indirect way to show low cost of our test. If DIRA assay is run in-house with our Sebia set-up it will cost around $130.
References:
- McCudden, C., et al., Monitoring multiple myeloma patients treated with daratumumab: teasing out monoclonal antibody interference. Clin Chem Lab Med, 2016. 54(6): p. 1095-104.
- Overdijk, M.B., et al., Antibody-mediated phagocytosis contributes to the anti-tumor activity of the therapeutic antibody daratumumab in lymphoma and multiple myeloma. MAbs, 2015. 7(2): p. 311-21.
Round 2
Reviewer 1 Report
The authors have addressed my comments and improved their manuscript in response to peer review. The supplementary figures added are very useful for readers. They seem to be slightly lower quality scans/slightly blurred compared perhaps compared to the other IFE figures used in the manuscript (maybe just harder to see as they are smaller?) but figure quality is an issue to be resolved with the journal. I have no further comments or changes.